# APPRENTICE: USING KNOWLEDGE DISTILLATION TECHNIQUES TO IMPROVE LOW-PRECISION NETWORK ACCURACY

**Asit Mishra & Debbie Marr**
Accelerator Architecture Lab
Intel Labs
{asit.k.mishra,debbie.marr}@intel.com

## ABSTRACT

Deep learning networks have achieved state-of-the-art accuracies on computer vision workloads like image classification and object detection. The performant systems, however, typically involve big models with numerous parameters. Once trained, a challenging aspect for such top performing models is deployment on resource constrained inference systems — the models (often deep networks or wide networks or both) are compute and memory intensive. *Low-precision numerics* and model compression using *knowledge distillation* are popular techniques to lower both the compute requirements and memory footprint of these deployed models. In this paper, we study *combination* of these two techniques and show that the performance of low-precision networks can be significantly improved by using knowledge distillation techniques. Our approach, $\mathcal{A}$pprentice, achieves state-of-the-art accuracies using ternary precision and 4-bit precision for variants of ResNet architecture on ImageNet dataset. We present three schemes using which one can apply knowledge distillation techniques to various stages of the train-and-deploy pipeline.

## 1 INTRODUCTION

**Background**: Today's high performing deep neural networks (DNNs) for computer vision applications comprise of multiple layers and involve numerous parameters. These networks have $\mathcal{O}$(Giga-FLOPS) compute requirements and generate models which are $\mathcal{O}$(Mega-Bytes) in storage (Canziani et al., 2016). Further, the memory and compute requirements during training and inference are quite different (Mishra et al., 2017). Training is performed on big datasets with large batch-sizes where memory footprint of activations dominates the model memory footprint. On the other hand, batch-size during inference is typically small and the model's memory footprint dominates the runtime memory requirements.

Because of complexity in compute, memory and storage requirements, training phase of the networks is performed on CPU and/or GPU clusters in a distributed computing environment. Once trained, a challenging aspect is deployment of trained models on resource constrained inference systems such as portable devices or sensor networks, and for applications in which real-time predictions are required. Performing inference on edge-devices comes with severe constraints on memory, compute and power. Additionally, ensemble based methods, which one can potentially use to get improved accuracy predictions, become prohibitive in resource constrained systems.

Quantization using low-precision numerics (Vanhoucke et al., 2011; Zhou et al., 2016; Lin et al., 2015; Miyashita et al., 2016; Gupta et al., 2015; Zhu et al., 2016; Rastegari et al., 2016; Courbariaux et al., 2015; Umuroglu et al., 2016; Mishra et al., 2017) and model compression (Buciluǎ et al., 2006; Hinton et al., 2015; Romero et al., 2014) have emerged as popular solutions for resource constrained deployment scenarios. With quantization, a low-precision version of network model is generated and deployed on the device. Operating in lower precision mode reduces compute as well as data movement and storage requirements. However, majority of existing works in low-precision DNNs sacrifice accuracy over baseline full-precision networks. With model compression, a smaller

low memory footprint network is trained to mimic the behaviour of the original complex network. During this training, a process called, knowledge distillation is used to "transfer knowledge" from the complex network to the smaller network. Work by Hinton et al. (2015) shows that the knowledge distillation scheme can yield networks at comparable or slightly better accuracy than the original complex model. However, to the best of our knowledge, all prior works using model compression techniques target compression at full-precision.

**Our proposal**: In this paper, we study the combination of network quantization with model compression and show that accuracies of low-precision networks can be significantly improved by using knowledge distillation techniques. Previous studies on model compression use a large network as the teacher network and a small network as the student network. The small student network learns from teacher network using distillation process. The network architecture of the student network is typically different from that of the teacher network – for e.g. Hinton et al. (2015) investigate a student network that has fewer number of neurons in the hidden layers compared to the teacher network. In our work, the student network has similar topology as that of teacher network, except that the student network has low-precision neurons compared to the teacher network which has neurons operating at full-precision.

We call our approach $\mathcal{A}$pprentice[1] and study three schemes which produce low-precision networks using knowledge distillation techniques. Each of these three schemes produce state-of-the-art ternary precision and 4-bit precision models.

In the first scheme, a low-precision network and a full-precision network are jointly trained from scratch using knowledge distillation scheme. Later in the paper we describe the rationale behind this approach. Using this scheme, a new state-of-the-art accuracy is obtained for ternary and 4-bit precision for ResNet-18, ResNet-34 and ResNet-50 on ImageNet dataset. In fact, using this scheme the accuracy of the full-precision model also slightly improves. This scheme then serves as the new baseline for the other two schemes we investigate.

In the second scheme, we start with a full-precision trained network and transfer knowledge from this trained network continuously to train a low-precision network from scratch. We find that the low-precision network converges faster (albeit to similar accuracies as the first scheme) when a trained complex network guides its training.

In the third scheme, we start with a trained full-precision large network and an apprentice network that has been initialised with full-precision weights. The apprentice network's precision is lowered and is fine-tuned using knowledge distillation techniques. We find that the low-precision network's accuracy marginally improves and surpasses the accuracy obtained via the first scheme. This scheme then sets the new state-of-the-art accuracies for the ResNet models at ternary and 4-bit precision.

Overall, the contributions of this paper are the techniques to obtain low-precision DNNs using knowledge distillation technique. Each of our scheme produces a low-precision model that surpasses the accuracy of the equivalent low-precision model published to date. One of our schemes also helps a low-precision model converge faster. We envision these accurate low-precision models to simplify the inference deployment process on resource constrained systems and even otherwise on cloud-based deployment systems.

## 2 MOTIVATION FOR LOW-PRECISION MODEL PARAMETERS

**Lowering precision of model parameters**: Resource constrained inference systems impose significant restrictions on memory, compute and power budget. With regard to storage, model (or weight) parameters and activation maps occupy memory during the inference phase of DNNs. During this phase memory is allocated for input (IFM) and output feature maps (OFM) required by a single layer in the DNN, and these dynamic memory allocations are reused for other layers. The total memory allocation during inference is then the maximum of IFM and maximum of OFM memory required across all the layers plus the sum of all weight tensors (Mishra et al., 2017). When inference phase for DNNs is performed with a small batch size, the memory footprint of the weights

---

[1]Dictionary defines apprentice as a person who is learning a trade from a skilled employer, having agreed to work for a fixed period at low wages. In our work, the apprentice is a low-precision network which is learning the knowledge of a high precision network (skilled employer) during a fixed number of epochs.

exceeds the footprint of the activation maps. This aspect is shown in Figure 1 for 4 different networks (AlexNet (Krizhevsky et al., 2012), Inception-Resnet-v2 (Szegedy et al., 2016), ResNet-50 and ResNet-101 (He et al., 2015)) running 224x224 image patches. Thus lowering the precision of the weight tensors helps lower the memory requirements during deployment. One other aspect of lowering memory footprint is that the working set size of the workload starts to fit on chip and by reducing accesses to DRAM (off-chip) memory, the compute core starts to see better performance and energy savings (DRAM accesses are expensive in latency and energy).

**Benefit of low-precision compute**: Low-precision compute simplifies hardware implementation. For example, the compute unit to perform the convolution operation (multiplication of two operands) involves a floating-point multiplier when using full-precision weights and activations. The floating-point multiplier can be replaced with a much simpler circuitry (xnor and popcount logic elements) when using binary precision for weights and activations (Courbariaux & Bengio, 2016; Rastegari et al., 2016; Courbariaux et al., 2015). Similarly, when using ternary precision for weights and full-precision for activations, the multiplier unit can be replaced with a sign comparator unit. Simpler hardware also helps lower the inference latency and energy budget. Thus, operating in lower precision mode reduces compute as well as data movement and storage requirements.

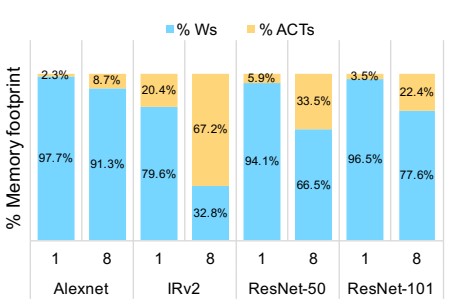

Figure 1: Memory footprint of activations (ACTs) and weights (W) during inference for mini-batch sizes 1 and 8.

The drawback of low-precision models, however, is degraded accuracy. We discuss later in the paper the network accuracies obtained using methods proposed in literature. These accuracies serve as the starting point and baselines we compare to in our work.

## 3   RELATED WORK

**Low-precision networks**: Low-precision DNNs are an active area of research. Most low-precision networks acknowledge the over parameterization aspect of today's DNN architectures and/or the aspect that lowering the precision of neurons post-training often does not impact the final performance. Reducing precision of weights for efficient inference pipeline has been very well studied. Works like Binary connect (BC) (Courbariaux et al., 2015), Ternary-weight networks (TWN) (Li & Liu, 2016), fine-grained ternary quantization (Mellempudi et al., 2017) and INQ (Zhou et al., 2017) target precision reduction of network weights. Accuracy is almost always affected when quantizing the weights significantly below 8-bits of precision. For AlexNet on ImageNet, TWN loses 5% Top-1 accuracy. Schemes like INQ, work in Sung et al. (2015) and Mellempudi et al. (2017) do fine-tuning to quantize the network weights.

Work in XNOR-NET (Rastegari et al., 2016), binary neural networks (Courbariaux & Bengio, 2016), DoReFa (Zhou et al., 2016) and trained ternary quantization (TTQ) (Zhu et al., 2016) target training pipeline. While TTQ targets weight quantization, most works targeting activation quantization show that quantizing activations always hurt accuracy. XNOR-NET approach degrades Top-1 accuracy by 12% and DoReFa by 8% when quantizing both weights and activations to 1-bit (for AlexNet on ImageNet). Work by Gupta et al. (2015) advocates for low-precision fixed-point numbers for training. They show 16-bits to be sufficient for training on CIFAR10 dataset. Work by Seide et al. (2014) quantizes gradients in a distributed computing system.

**Knowledge distillation methods**: The general technique in distillation based methods involves using a teacher-student strategy, where a large deep network trained for a given task teaches shallower student network(s) on the same task. The core concepts behind knowledge distillation or transfer technique have been around for a while. Buciluǎ et al. (2006) show that one can compress the information in an ensemble into a single network. Ba & Caurana (2013) extend this approach to study shallow, but wide, fully connected topologies by mimicking deep neural networks. To facil-

itate learning, the authors introduce the concepts of learning on logits rather than the probability distribution.

Hinton et al. (2015) propose a framework to transfer knowledge by introducing the concept of temperature. The key idea is to divide the logits by a temperature factor before performing a Softmax function. By using a higher temperature factor the activations of incorrect classes are boosted. This then facilitates more information flowing to the model parameters during back-propagation operation. FitNets (Romero et al., 2014) extend this work by using intermediate hidden layer outputs as target values for training a deeper, but thinner, student model. Net2Net (Chen et al., 2015a) also uses a teacher-student network system with a function-preserving transformation approach to initialize the parameters of the student network. The goal in Net2Net approach is to accelerate the training of a larger student network. Zagoruyko & Komodakis (2016) use attention as a mechanism for transferring knowledge from one network to another. In a similar theme, Yim et al. (2017) propose an information metric using which a teacher DNN can transfer the distilled knowledge to other student DNNs. In N2N learning work, Ashok et al. (2017) propose a reinforcement learning based approach for compressing a teacher network into an equally capable student network. They achieve a compression factor of 10x for ResNet-34 on CIFAR datasets.

**Sparsity and hashing**: Few other popular techniques for model compression are pruning (LeCun et al., 1990; Han et al., 2015a; Wen et al., 2016; Han et al., 2015b), hashing (Weinberger et al., 2009) and weight sharing (Chen et al., 2015b; Denil et al., 2013). Pruning leads to removing neurons entirely from the final trained model making the model a sparse structure. With hashing and weight sharing schemes a hash function is used to alias several weight parameters into few hash buckets, effectively lowering the parameter memory footprint. To realize benefits of sparsity and hashing schemes during runtime, efficient hardware support is required (e.g. support for irregular memory accesses (Han et al., 2016; Venkatesh et al., 2016; Parashar et al., 2017)).

## 4 KNOWLEDGE DISTILLATION

We introduce the concept of knowledge distillation in this section. Buciluǎ et al. (2006), Hinton et al. (2015) and Urban et al. (2016) analyze this topic in great detail.

Figure 2 shows the schematic of the knowledge distillation setup. Given an input image $x$, a teacher DNN maps this image to predictions $p^T$. The $C$ class predictions are obtained by applying Softmax function on the un-normalized log probability values $z$ (the logits), i.e. $p^T = e^{z_k^T}/\sum_j^C e^{z_j^T}$. The same image is fed to the student network and it predicts $p^A = e^{z_k^A}/\sum_j^C e^{z_j^A}$. During training, the cost function, $\mathcal{L}$, is given as:

$$\mathcal{L}(x; W_T, W_A) = \alpha \mathcal{H}(y, p^T) + \beta \mathcal{H}(y, p^A) + \gamma \mathcal{H}(z^T, p^A) \tag{1}$$

where, $W_T$ and $W_A$ are the parameters of the teacher and the student (apprentice) network, respectively, $y$ is the ground truth, $\mathcal{H}(\cdot)$ denotes a loss function and, $\alpha$, $\beta$ and $\gamma$ are weighting factors to prioritize the output of a certain loss function over the other.

In equation 1, lowering the first term of the cost function gives a better teacher network and lowering the second term gives a better student network. The third term is the knowledge distillation term whereby the student network attempts to mimic the knowledge in the teacher network. In Hinton et al. (2015), the logits of the teacher network are divided by a temperature factor $\tau$. Using a higher value for $\tau$ produces a softer probability distribution when taking the Softmax of the logits. In our studies, we use cross-entropy function for $\mathcal{H}(\cdot)$, set $\alpha = 1$, $\beta = 0.5$ and $\gamma = 0.5$ and, perform the transfer learning process using the logits (inputs to the Softmax function) of the teacher network. In our experiments we study the effect of varying the depth of the teacher and the student network, and the precision of the neurons in the student network.

## 5 OUR APPROACH - APPRENTICE NETWORK

Low-precision DNNs target the storage and compute efficiency aspects of the network. Model compression targets the same efficiency parameters from the point of view of network architecture. With

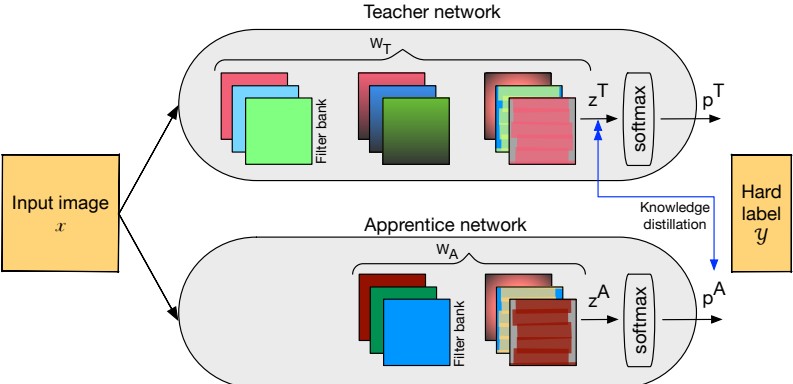

Figure 2: Schematic of the knowledge distillation setup. The teacher network is a high precision network and the apprentice network is a low-precision network.

$\mathcal{A}$pprentice we combine both these techniques to improve the network accuracy as well as the runtime efficiency of DNNs. Using the teacher-student setup described in the last section, we investigate three schemes using which one can obtain a low-precision model for the student network. The first scheme (scheme-A) jointly trains both the networks - full-precision teacher and low-precision student network. The second scheme (scheme-B) trains only the low-precision student network but distills knowledge from a trained full-precision teacher network throughout the training process. The third scheme (scheme-C) starts with a trained full-precision teacher and a full-precision student network but fine-tunes the student network after lowering its precision. Before we get into the details of each of these schemes, we discuss the accuracy numbers obtained using low-precision schemes described in literature. These accuracy figures serve as the baseline for comparative analysis.

## 5.1 TOP-1 ERROR WITH PRIOR PROPOSALS FOR LOW-PRECISION NETWORKS

We focus on sub 8-bits precision for inference deployments, specifically ternary and 4-bits precision. We found TTQ (Zhu et al., 2016) scheme achieving the state-of-the-art accuracy with ternary precision for weights and full-precision (32-bits floating-point) for activations. On Imagenet-1K (Russakovsky et al., 2015), TTQ achieves 33.4% Top-1 error rate with a ResNet-18 model. We implemented TTQ scheme for ResNet-34 and ResNet-50 models trained on Imagenet-1K and achieved 28.3% and 25.6% Top-1 error rates, respectively. This scheme is our baseline for 2-bits weight and full-precision activations. For 2-bits weight and 8-bits activation, we find work by Mellempudi et al. (2017) to achieve the best accuracies reported in literature. For ResNet-50, Mellempudi et al. (2017) obtain 29.24% Top-1 error. We consider this work to be our baseline for 2-bits weight and 8-bits activation models.

For 4-bits precision, we find WRPN scheme (Mishra et al., 2017) to report the highest accuracy. We implemented this scheme for 4-bits weight and 8-bits activations. For ResNet-34 and ResNet-50 models trained on Imagenet-1K, we achieve 29.7% and 28.4% Top-1 error rates, respectively.

## 5.2 SCHEME-A: JOINT TRAINING OF TEACHER-STUDENT NETWORKS

In the first scheme that we investigate, a full-precision teacher network is jointly trained with a low-precision student network. Figure 2 shows the overall training framework. We use ResNet topology for both the teacher and student network. When using a certain depth for the student network, we pick the teacher network to have either the same or larger depth.

In Buciluǎ et al. (2006) and Hinton et al. (2015), only the student network trains while distilling knowledge from the teacher network. In our case, we jointly train with the rationale that the teacher network would continuously guide the student network not only with the final trained logits, but also on what path the teacher takes towards generating those final higher accuracy logits.

We implement pre-activation version of ResNet (He et al., 2016) in TensorFlow (Abadi et al., 2015). The training process closely follows the recipe mentioned in Torch implementation of ResNet - we use a batch size of 256 and no hyper-parameters are changed from what is mentioned in the recipe. For the teacher network, we experiment with ResNet-34, ResNet-50 and ResNet-101 as options. For the student network, we experiment with low-precision variants of ResNet-18, ResNet-34 and ResNet-50.

For low-precision numerics, when using ternary precision we use the ternary weight network scheme (Li & Liu, 2016) where the weight tensors are quantized into $\{-1, 0, 1\}$ with a per-layer scaling coefficient computed based on the mean of the positive terms in the weight tensor. We use the WRPN scheme (Mishra et al., 2017) to quantize weights and activations to 4-bits or 8-bits. We do not lower the precision of the first layer and the final layer in the apprentice network. This is based on the observation in almost all prior works that lowering the precision of these layers degrades the accuracy dramatically. While training and during fine-tuning, the gradients are still maintained at full-precision.

Table 1: Top-1 validation set error rate (%) on ImageNet-1K for ResNet-18 student network as precision of activations (A) and weight (W) changes. The last three columns show error rate when the student ResNet-18 is paired with ResNet-34, ResNet-50 and ResNet-101.

|  | ResNet-18 Baseline | ResNet-18 with ResNet-34 | ResNet-18 with ResNet-50 | ResNet-18 with ResNet-101 |
|---|---|---|---|---|
| 32A, 32W | 30.4 | 30.2 | 30.1 | 30.1 |
| 32A, 2W | 33.4 | 31.7 | 31.5 | 31.8 |
| 8A, 4W | 33.6 | 29.6 | 29.6 | 29.9 |
| 8A, 2W | 33.9 | 32.0 | 32.2 | 32.4 |

**Results with ResNet-18**: Table 1 shows the effect of lowering precision on the accuracy (Top-1 error) of ResNet-18 with baseline (no teacher) and with ResNet-34, ResNet-50 and ResNet-101 as teachers. In the table, $A$ denotes the precision of the activation maps (in bits) and $W$ denotes the precision of the weights. The baseline Top-1 error for full-precision ResNet-18 is 30.4%. By lowering the precision without using any help from a teacher network, the accuracy drops by 3.5% when using ternary and 4-bits precision (the column corresponding to "Res-18 Baseline" in the table). With distillation based technique, the accuracy of low-precision configurations improves significantly. In fact, the accuracy of the full-precision ResNet-18 also improves when paired with a larger full-precision ResNet model (the row corresponding to "32A, 32W" in Table 1). The best full-precision accuracy was achieved with a student ResNet-18 and ResNet-101 as the teacher (improvement by 0.35% over the baseline). The gap between full-precision ResNet-18 and the best achieved ternary weight ResNet-18 is only 1% (improvement of 2% over previous best). With "8A, 4W", we find the accuracy of the student ResNet-18 model to beat the baseline accuracy. We hypothesize regularization with low-precision (and distillation) to be the reason for this. "8A, 4W" improving the accuracy beyond baseline figure is only seen for ResNet-18.

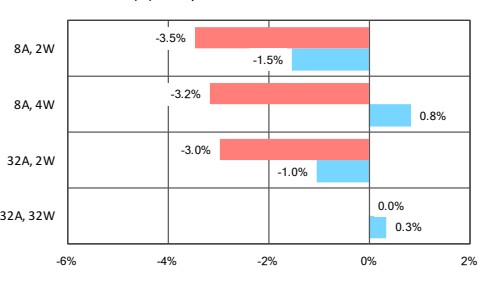

Figure 3: Difference in Top-1 error rate for low-precision variants of ResNet-18 with (blue bars) and without (red bars) distillation scheme. The difference is calculated from the accuracy of ResNet-18 with full-precision numerics. Higher % difference denotes a better network configuration.

Figure 3 shows the *difference* in Top-1 error rate achieved by our best low-precision student networks (when trained under the guidance of a teacher network) versus not using any help from a teacher network. For this figure, the difference in Top-1 error of the best low-precision student network is calculated from the baseline full-precision network (i.e. ResNet-18 with 30.4% Top-1 error), i.e. we want to see how close a low-precision student network can come to a full-precision baseline model.

We find our low-precision network accuracies to significantly close the gap between full-precision accuracy (and for some configurations even beat the baseline accuracy).

Hinton et al. (2015) mention improving the baseline full-precision accuracy when a student network is paired with a teacher network. They mention improving the accuracy of a small model on MNIST dataset. We show the efficacy of distillation based techniques on a much bigger model (ResNet) with much larger dataset (ImageNet).

Table 2: Top-1 validation set error rate (%) on ImageNet-1K for ResNet-34 student network as precision of activations (A) and weight (W) changes. The last three columns show error rate when the student ResNet-34 is paired with ResNet-34, ResNet-50 and ResNet-101.

|  | ResNet-34 Baseline | ResNet-34 with ResNet-34 | ResNet-34 with ResNet-50 | ResNet-34 with ResNet-101 |
|---|---|---|---|---|
| 32A, 32W | 26.4 | 26.3 | 26.1 | 26.1 |
| 32A, 2W | 28.3 | 27.6 | 27.2 | 27.2 |
| 8A, 4W | 29.7 | 27.0 | 26.9 | 26.9 |
| 8A, 2W | 30.8 | 28.8 | 28.8 | 28.5 |

Table 3: Top-1 validation set error rate (%) on ImageNet-1K for ResNet-50 student network as precision of activations (A) and weight (W) changes. The final two columns show error rate when the student ResNet-50 is paired with ResNet-50 and ResNet-101.

|  | ResNet-50 Baseline | ResNet-50 with ResNet-50 | ResNet-50 with ResNet-101 |
|---|---|---|---|
| 32A, 32W | 23.8 | 23.7 | 23.5 |
| 32A, 2W | 26.1 | 25.4 | 25.3 |
| 8A, 4W | 28.5 | 25.5 | 25.3 |
| 8A, 2W | 29.2 | 27.3 | 27.2 |

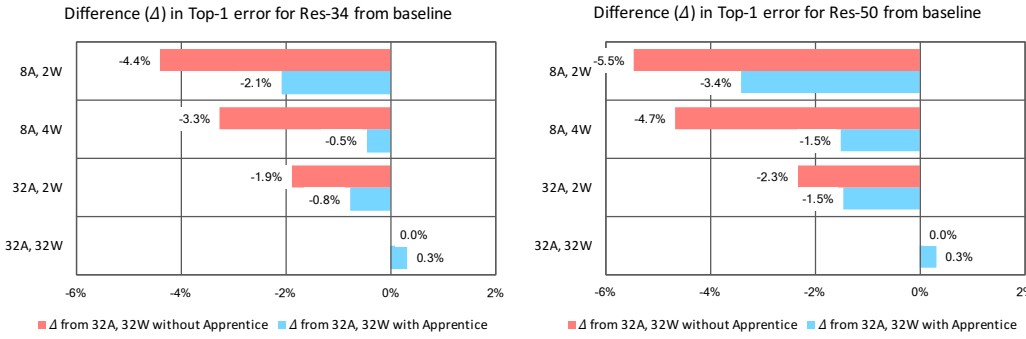

(a) Apprentice versus baseline accuracy for ResNet-34. (b) Apprentice versus baseline accuracy for ResNet-50.

Figure 4: Difference in Top-1 error rate for low-precision variants of ResNet-34 and ResNet-50 with (blue bars) and without (red bars) distillation scheme. The difference is calculated from the accuracy of the baseline network (ResNet-34 for (a) and ResNet-50 for (b)) operating at full-precision. Higher % difference denotes a better network configuration.

**Results with ResNet-34 and ResNet-50**: Table 2 and Table 3 show the effect of lowering precision on the accuracy of ResNet-34 and ResNet-50, respectively, with distillation based technique. With a student ResNet-34 network, we use ResNet-34, ResNet-50 and ResNet-101 as teachers. With a student ResNet-50 network, we use ResNet-50 and ResNet-101 as teachers. The Top-1 error for full-precision ResNet-34 is 26.4%. Our best 4-bits weight and 8-bits activation ResNet-34 is within 0.5% of this number (26.9% error rate with ResNet-34 student and ResNet-50 teacher). This

significantly improves upon the previously reported error rate of 29.7%. 4-bits weight and 8-bits activation for ResNet-50 gives us a model that is within 1.5% of full-precision model accuracy (25.3% vs. 23.8%). Figure 4a and Figure 4b show the difference in Top-1 error achieved by our best low-precision ResNet-34 and ResNet-50 student networks, respectively, and compares with results obtained using methods proposed in literature. Our $\mathcal{A}$pprentice scheme significantly closes the gap between full-precision baseline networks and low-precision variants of the same networks. In most cases we see our scheme to better the previously reported accuracy numbers by 1.5%-3%.

**Discussion**: In scheme-A, we use a teacher network that is always as large or larger in number of parameters than the student network. We experimented with a ternary ResNet-34 student network which was paired with a full-precision ResNet-18. The ternary model for ResNet-34 is about 8.5x smaller in size compared to the full-precision ResNet-18 model. The final trained accuracy of the ResNet-34 ternary model with this setup is 2.7% worse than that obtained by pairing the ternary ResNet-34 network with a ResNet-50 teacher network. This suggests that the distillation scheme works only when the teacher network is higher in accuracy than the student network (and not necessarily bigger in capacity). Further, the benefit from using a larger teacher network saturates at some point. This can be seen by picking up a precision point, say "32A, 2W" and looking at the error rates along the row in Table 1, 2 and 3.

One concern, we had in the early stages of our investigation, with joint training of a low-precision small network and a high precision large network was the influence of the small network's accuracy on the accuracy of the large network. When using the joint cost function, the smaller network's probability scores are matched with the predictions from the teacher network. The joint cost is added as a term to the total loss function (equation 1). This led us to posit that the larger network's learning capability will be affected by the inherent impairment in the smaller low-precision network. Further, since the smaller student network learns form the larger teacher network, a vicious cycle might form where the student network's accuracy will further drop because the teacher network's learning capability is being impeded. However, in practice, we did not see this phenomenon occurring - in each case where the teacher network was jointly trained with a student network, the accuracy of the teacher network was always within 0.1% to 0.2% of the accuracy of the teacher network without it jointly supervising a student network. This could be because of our choice of $\alpha$, $\beta$ and $\gamma$ values.

In Section 4, we mentioned about temperature, $\tau$, for Softmax function and hyper-parameters $\alpha = 1$, $\beta = 0.5$ and $\gamma = 0.5$. Since, we train directly on the logits of the teacher network, we did not have to experiment with the appropriate value of $\tau$. $\tau$ is required when training on the soft targets produced by the teacher network. Although we did not do extensive studies experimenting with training on soft targets as opposed to logits, we find that $\tau = 1$ gives us best results when training on soft targets. Hinton et al. (2015) mention that when the student network is significantly smaller than the teacher network, small values of $\tau$ are more effective than large values. For few of the low-precision configurations, we experimented with $\alpha = \beta = \gamma = 1$, and, $\alpha = 0.9$, $\beta = 1$ and $\gamma = 0.1$ or $0.3$. Each of these configurations, yielded a lower performance model compared to our original choice for these parameters.

For the third term in equation 1, we experimented with a mean-squared error loss function and also a loss function with logits from both the student and the teacher network (i.e. $\mathcal{H}(z^T, z^A)$). We did not find any improvement in accuracy compared to our original choice of the cost function formulation. A thorough investigation of the behavior of the networks with other values of hyper-parameters and different loss functions is an agenda for our future work.

Overall, we find the distillation process to be quite effective in getting us high accuracy low-precision models. *All our low-precision models surpass previously reported low-precision accuracy figures.* For example, TTQ scheme achieves 33.4% Top-1 error rate for ResNet-18 with 2-bits weight. Our best ResNet-18 model, using scheme-A, with 2-bits weight achieves ∼31.5% error rate, improving the model accuracy by ∼2% over TTQ. Similarly, the scheme in Mellempudi et al. (2017) achieves 29.2% Top-1 error with 2-bits weight and 8-bits activation. The best performing $\mathcal{A}$pprentice network at this precision achieves 27.2% Top-1 error. For Scheme-B and Scheme-C, which we describe next, Scheme-A serves as the new baseline.

## 5.3 SCHEME-B: DISTILLING KNOWLEDGE FROM A TEACHER

In this scheme, we start with a trained teacher network. Referring back to Figure 2, the input image is passed to both the teacher and the student network, except that the learning with back-propagation happens only in the low precision student network which is trained from scratch. This is the scheme used by Bucilă et al. (2006) and Hinton et al. (2015) for training their student networks. In this scheme, the first term in equation 1 zeroes out and only the last two terms in the equation contribute toward the loss function.

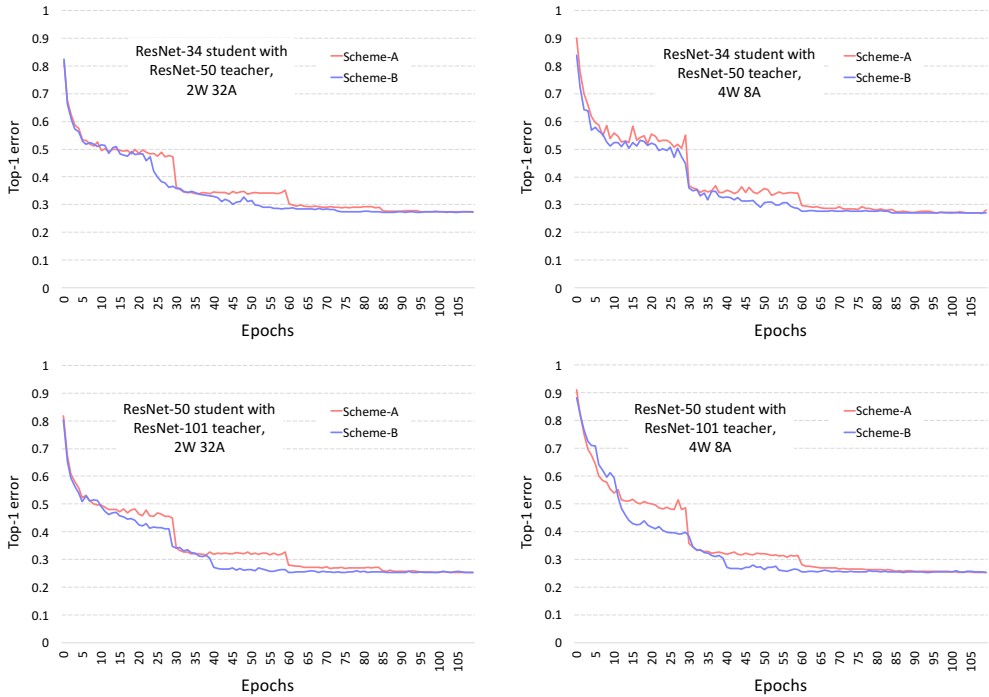

Figure 5: Top-1 error rate versus epochs of four student networks using scheme-A and scheme-B.

With scheme-B, one can pre-compute and store the logit values for the input images on disk and access them during training the student network. This saves the forward pass computations in the teacher network. Scheme-B might also help the scenario where a student network attempts to learn the "dark knowledge" from a teacher network that has already been trained on some private or sensitive data (in addition to the data the student network is interested in training on).

With scheme-A, we had the hypothesis that the student network would be influenced by not only the "dark knowledge" in the teacher network but also the path the teacher adopts to learn the knowledge. With scheme-B we find, that the student network gets to similar accuracy numbers as the teacher network albeit at fewer number of epochs.

With this scheme, the training accuracies are similar to that reported in Table 1, 2 and 3. The low-precision student networks, however, learn in fewer number of epochs. Figure 5 plots the Top-1 error rates for few of the configurations from our experiment suite. In each of these plots, the student network in scheme-B converges around 80th-85th epoch compared to about 105 epochs in scheme-A. In general, we find the student networks with scheme-B to learn in about 10%-20% fewer epochs than the student networks trained using scheme-A.

## 5.4 SCHEME-C: FINE-TUNING THE STUDENT MODEL

Scheme-C is very similar to scheme-B, except that the student network is primed with full precision training weights before the start of the training process. At the beginning of the training process, the weights and activations are lowered and the student network is sort of fine-tuned on the dataset.

Similar to scheme-B, only the final two terms in equation 1 comprise the loss function and the low-precision student network is trained with back-propagation algorithm. Since, the network starts from a good initial point, comparatively low learning rate is used throughout the training process. There is no clear recipe for learning rates (and change of learning rate with epochs) which works across all the configurations. In general, we find training with a learning rate of 1e-3 for 10 to 15 epochs, followed by 1e-4 for another 5 to 10 epochs, followed by 1e-5 for another 5 epochs to give us the best accuracy. Some configurations run for about 40 to 50 epochs before stabilizing. For these configurations, we find training using scheme-B with warm startup (train the student network at full-precision for about 25-30 epochs before lowering the precision) to be equally good. Wu (2016) investigate a similar scheme for binary precision on AlexNet. Our experiments show that distillation is an overkill for AlexNet and one can get comparable accuracies using techniques proposed in (Tang et al., 2017; Mishra et al., 2017). Further, Wu (2016) hypothesize that distillation scheme will work on larger networks, we show in this paper how to make it work. Tann et al. (2017) use a similar scheme for AlexNet and mention starting from a non-global optimal checkpoint gives better accuracy, though we did not find this observation to hold in our experiments.

We find the final accuracy of the models obtained using scheme-C to be (marginally) better than those obtained using scheme-A or scheme-B. Table 4 shows error rates of few configurations of low-precision student network obtained using scheme-A (or scheme-B) and scheme-C. For ResNet-50 student network, the accuracy with ternary weights is further improved by 0.6% compared to that obtained using scheme-A. Note that the performance of ternary networks obtained using scheme-A are already state-of-the-art. Hence, for ResNet-50 ternary networks, 24.7% Top-1 error rate is the new state-of-the-art. With this, ternary ResNet-50 is within 0.9% of baseline accuracy (23.8% vs. 24.7%). Similarly, with 4-bits weight and 8-bits activations, ResNet-50 model obtained using scheme-C is 0.4% better than that obtained with scheme-A (closing the gap to be within 1.3% of full-precision ResNet-50 accuracy).

Table 4: Top-1 ImageNet-1K validation set error rate (%) with scheme-A and scheme-C for ResNet-34 and ResNet-50 student networks with ternary and 4-bits precision.

|  | 32A, 2W with scheme-A or B | 32A, 2W with scheme-C |
|---|---|---|
| ResNet-34 student with ResNet-50 teacher | 27.2 | 26.9 |
| ResNet-50 student with ResNet101 teacher | 25.3 | 24.7 |
|  | 8A, 4W with scheme-A or B | 8A, 4W with scheme-C |
| ResNet-34 student with ResNet-50 teacher | 26.9 | 26.8 |
| ResNet-50 student with ResNet101 teacher | 25.5 | 25.1 |

Scheme-C is useful when one already has a trained network which can be fine-tuned using knowledge distillation schemes to produce a low-precision variant of the trained network.

## 5.5 Discussion - ternary precision versus sparsity

As mentioned earlier, low-precision is a form of model compression. There are many works which target network sparsification and pruning techniques to compress a model. With ternary precision models, the model size reduces by a factor of $2/32$ compared to full-precision models. With $\mathcal{A}$pprentice, we show how one can get a performant model with ternary precision. Many works targeting network pruning and sparsification target a full-precision model to implement their scheme. To be comparable in model size to ternary networks, a full-precision model needs to be sparsified by 93.75%. Further, to be effective, a sparse model needs to store a key for every non-zero value denoting the position of the value in the weight tensor. This adds storage overhead and a sparse model needs to be about 95% sparse to be at-par in memory size as a 2-bit model. Note that ternary precision also has inherent sparsity (zero is a term in the ternary symbol dictionary) – we find our ternary models to be about 50% sparse. In work by Wen et al. (2016) and Han et al. (2015b), sparsification of full-precision networks is proposed but the sparsity achieved is less than 93.75%. Further, the network accuracy using techniques in both these works lead to larger degradation in accuracy compared to our ternary models. Overall, we believe, our ternary precision models to be state-of-the-art

not only in accuracy (we better the accuracy compared to prior ternary precision models) but also when one considers the size of the model at the accuracy level achieved by low-precision or sparse networks.

# 6 CONCLUSIONS

While low-precision networks have system-level benefits, the drawback of such models is degraded accuracy when compared to full-precision models. We present three schemes based on knowledge distillation concept to improve the accuracy of low-precision networks and close the gap between the accuracy of these models and full-precision models. Each of the three schemes improve the accuracy of the low-precision network configuration compared to prior proposals. We motivate the need for a smaller model size in low batch, real-time and resource constrained inference deployment systems. We envision the low-precision models produced by our schemes to simplify the inference deployment process on resource constrained systems and on cloud-based deployment systems where low latency is a critical requirement.

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

# 7 APPENDIX: ANALYSIS WITH RESNET ON CIFAR-10 DATASET

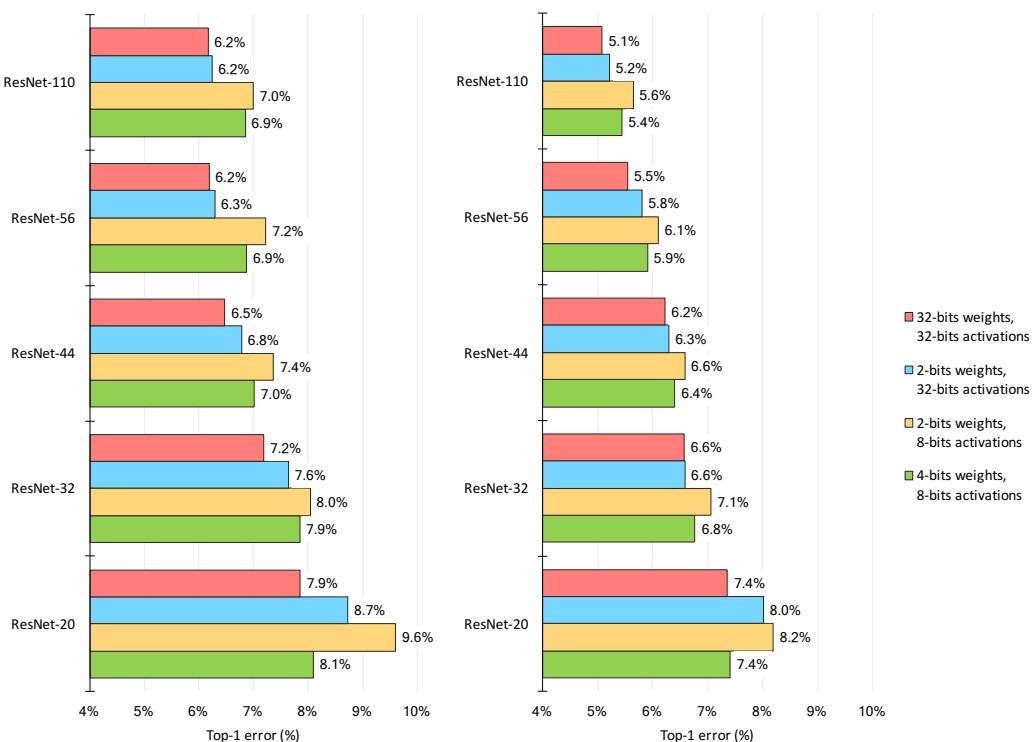

(a) Top-1 error without Apprentice scheme.   (b) Top-1 error *using* Apprentice scheme-A.

Figure 6: Comparison of various configurations of ResNet on CIFAR-10 with and without $\mathcal{A}$pprentice scheme.

In addition to ImageNet dataset, we also experiment with $\mathcal{A}$pprentice scheme on CIFAR-10 dataset. CIFAR-10 dataset (Krizhevsky, 2009) consists of 50K training images and 10K testing images in 10 classes. We use various depths of ResNet topology for this study. Our implementation of ResNet for CIFAR-10 closely follows the configuration in He et al. (2015). The network inputs are 32×32 images. The first layer is a 3×3 convolutional layer followed by a stack of $6n$ layers with 3×3 convolutions on feature map sizes 32, 16 and 8; with $2n$ layers for each feature map size. The numbers of filters are 16, 32 and 64 in each set of $2n$ layers. This is followed by a global average pooling, a 10-way fully connected layer and a softmax layer. Thus, in total there are $6n+2$ weight layers.

Figure 6a shows the impact of lowering precision as the depth of ResNet varies. As the network becomes larger in size, the impact of lowering precision is diminished (relative to the accuracy of the network at that depth when using full-precision). For example, with ResNet-110, full-precision Top-1 error rate is 6.19%. At the same depth, ternarizing the model gives similar error rate (6.24%). Comparing this with ResNet-20, the gap between full-precision and ternary model (2-bits weight and 32-bits activations) is 0.8% (7.9% vs. 8.7% Top-1 error). Overall, we find that ternarizing a model closely follows accuracy of baseline full-precision model. However, lowering both weights and activations almost always leads to large accuracy degradation. Accuracy of 2-bits weight and 8-bits activation network is 0.8%-1.6% worse than full-precision model. Using $\mathcal{A}$pprentice scheme this gap is considerably lowered.

Figure 6b shows the impact of lowering precision when a low-precision (student) network is paired with a full-precision (teacher) network. For this analysis we use scheme-A where we jointly train both the teacher and student network. The mix of ResNet depths we used for this study are ResNet-20, 32, 44, 56, 110 and 182. ResNet-20 student network was paired with deeper ResNets from this mix, i.e. ResNet-32, 44, 56, 110 and 182 (as five separate experiments). Similarly, ResNet-44

student network was paired with deeper ResNet-56 and 110 (as two different set of experiments). ResNet-110 student network used ResNet-182 as its teacher network. For a particular ResNet depth, the figure plots the minimum error rate across each of the experiments.

*We find* $\mathcal{A}$pprentice *scheme to improve the baseline full-precision accuracy*. The scheme also helps close the gap between the *new improved baseline accuracy* and the accuracy when lowering the precision of the weights and activations. The gap between 2-bits weight and 8-bits activation network is now 0.4%-0.8% worse than full-precision model. With ImageNet dataset, the accuracy of full-precision networks also improves but very marginally (by 0.3%). However, the impact of distillation technique on CIFAR-10 is quite pronounced - for example, top-1 error lowers by 1.1% for ResNet-110.

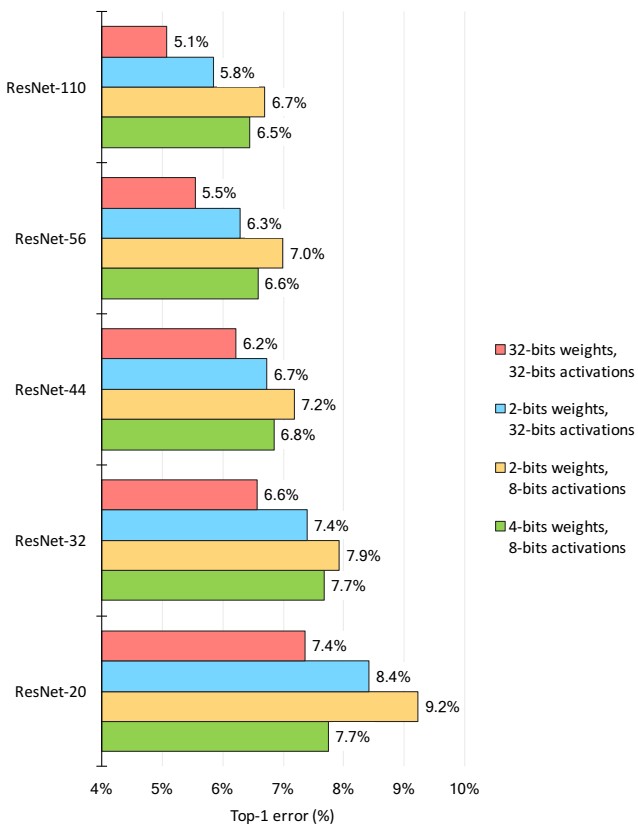

Figure 7: Distillation followed by quantization.

We experimented with a variation of scheme-C where a network is first compressed using distillation scheme (using a deeper ResNet as the teacher network) followed by lowering the precision and fine-tuning. The fine-tuning is done for 35-40 epochs with a very low learning-rate without the influence of any teacher network (no distillation). For this experiment, the student network starts from a higher accuracy compressed model compared to scheme-C (since distillation improves accuracy of student network at full-precision as well). Figure 7 shows the results with this experimental setting. For each configuration, we find the error-rate to lie in between the error-rates shown in Figure 6a and Figure 6b for the corresponding configuration, i.e. this scheme is better than low-precision training from scratch but not as good as training with methodology described in scheme-A. On an average, we find scheme-A to give 0.7% better accuracy at low-precision configurations compared to the scheme mentioned here highlighting the benefits of "joint" low-precision training from scratch with distillation ($\mathcal{A}$pprentice scheme). Many works proposing low-precision knobs advocate for training from scratch or training (for a significant number of epochs) with warm-startup – the conclusions from this experiment are in line with the observations in these papers.

## 8  FUTURE RESEARCH

Some works proposing low-precision networks advocate for making the layers wider (or the model larger) to recover accuracy at low-precision. These works propose making the layers wider by 2x or 3x. While these works show the benefits of low-precision, making the model larger increases the number of raw computations. Future work could investigate low-precision and less layer widening factor (say 1.10x or 1.25x). This would help inference latency while maintaining accuracy at-par with baseline full-precision networks.

As mentioned in section 5.5, sparsifying a model more than a certain percentage leads to accuracy loss. Investigating hyper-sparse network models without accuracy loss using distillation based schemes is another interesting avenue of further research.

