# OpenReview forum: "Apprentice: Using Knowledge Distillation Techniques To Improve Low-Precision Network Accuracy"
_ICLR.cc/2018/Conference — Accept (Poster)_

### Official Review · AnonReviewer1 · 2017-11-23
**speed & memory gains, transferability**

**Rating:** 7
**Confidence:** 4

**Review:**

The authors investigate knowledge distillation as a way to learn low precision networks. They propose three training schemes to train a low precision student network from a teacher network. They conduct experiments on ImageNet-1k with variants of ResNets and multiple low precision regimes and compare performance with previous works

Pros:
(+) The paper is well written, the schemes are well explained
(+) Ablations are thorough and comparisons are fair
Cons:
(-) The gap with full precision models is still large
(-) Transferability of the learned low precision models to other tasks is not discussed

The authors tackle a very important problem, the one of learning low precision models without comprosiming performance. For scheme-A, the authors show the performance of the student network under many low precision regimes and different depths of teacher networks. One observation not discussed by the authors is that the performance of the student network under each low precision regime doesn't improve with deeper teacher networks (see Table 1, 2 & 3). As a matter of fact, under some scenarios performance even decreases.

The authors do not discuss the gains of their best low-precision regime in terms of computation and memory.

Finally, the true applications for models with a low memory footprint are not necessarily related to image classification models (e.g. ImageNet-1k). How good are the low-precision models trained by the authors at transferring to other tasks? Is it possible to transfer student-teacher training practices to other tasks?

---

> ### Author Response · Authors · 2017-12-05
> **Response to "Cons" reviews and questions**
>
> Thank you for the thorough reviews.
>
> We defend the "Cons" reviews below:
>
> Gap from full precision: Agreed. However, compared to prior works the improvement is significant. For example, now with 8-bits activations and 4-bits weight, ResNet-34 without any change in network architecture is only 0.5% off from baseline full precision. This is currently the best Top-1 figure at this precision knob -- the best figure with prior techniques gave 3.3% degradation (so 2.8% improvement with our scheme). We believe that by making the models slightly larger (by 10% or so) we can close the gap between low-precision and full-precision networks -- this is our future work.
>
> Transferability: We believe, this aspect should not change with our scheme for low-precision settings -- we simply better the accuracy of a given network at low-precision (compared to prior proposals). However much the network was useful in transfer learning scenarios with low-precision tensors before, the network right now with our scheme would be similarly useful (if at all better when compared to prior works since we achieve a better accuracy with low-precision).
>
> Your question: Is it possible to transfer student-teacher training practices to other tasks?
> Although we did not focus on this aspect in this paper, we found the following 3 works (not an exhaustive list) that use the student-teacher training procedure for other deep-learning domains:
>
> 1. Transferring Knowledge from a RNN to a DNN, William Chan et al., ArXiv pre-print, 2015.
> 2. Recurrent Neural Network Training With Dark Knowledge Transfer, Zhiyuan Tang et al., ArXiv pre-print, 2016.
> 3. Simultaneous Deep Transfer Across Domains and Tasks, Eric Tzeng et al., ICCV 2015.
>
>
> The observation that accuracy does not improve when using bigger teacher network(s): We allude to this in Discussion part of Section 5.2 (page-8). We mention that the accuracy improvement saturates at some point. We will elaborate on this aspect in the final version of the paper.
>
> We will also discuss the merits of low precision on savings in compute and memory. We briefly discuss these aspects in Section 2 where we mention about simplification in hardware support required for inference. We will elaborate on these aspects and provide quantification in compute and memory footprint savings vs. accuracy in our final version of the paper.

---

### Official Review · AnonReviewer2 · 2017-11-27
**Interesting results but very limited contribution**

**Rating:** 7
**Confidence:** 4

**Review:**

The paper aims at improving the accuracy of a low precision network based on knowledge distillation from a full-precision network. Instead of distillation from a pre-trained network, the paper proposes to train both teacher and student network jointly. The paper shows an interesting result that the distilled low precision network actually performs better than high precision network.

I found the paper interesting but the contribution seems quite limited.

Pros:
1. The paper is well written and easy to read.
2. The paper reported some interesting result such as that the distilled low precision network actually performs better than high precision network, and that training jointly outperforms the traditional distillation method (fixing the teacher network) marginally.

Cons:
1. The name Apprentice seems a bit confusing with apprenticeship learning.
2. The experiments might be further improved by providing a systematic study about the effect of precisions in this work (e.g., producing more samples of precisions on activations and weights).
3. It is unclear how the proposed method outperforms other methods based on fine-tuning. It is also quite possible that after fine-tuning the compressed model usually performs quite similarly to the original model.

---

> ### Author Response · Authors · 2017-12-05
> **Defending the contributions and response to other questions**
>
> Thank you for the reviews. They are useful.
>
> We defend our contribution aspect below:
>
> Contributions: Distillation along with lowering precision has not been studied before. We show benefits of this combined approach - both distillation and low precision target model compression aspect - but when combined the benefits are significant. We also show how one can use the combined distillation and lowering precision approach to training as well as fine-tuning.
>
> Our approach achieves state-of-the-art in accuracy over prior proposals and using our approach we significantly close the gap between full-precision and low-precision model accuracy. We demonstrate the benefits on ImageNet with large networks (ResNet). For example, with ResNet-50 on ImageNet, prior work showed 4.7% accuracy degradation with 8-bits activation and 4-bits weight. We lower this gap to less than 1.5%.
>
> We believe ours to be the first work that targets both model compression (using knowledge distillation) and low-precision.
>
>
> Response to the Cons aspects:
> 1. We probably did not do a good job describing why we call our approach Apprentice. We will fix this and disambiguate from apprenticeship-based learning schemes.
>
> 2. The reason we focus on sub 8-bit precision is that model inference with 8-bits is becoming mainstream and we seek to target next-gen hardware architectures. Also, from a hardware point-of-view 8-bits, 4-bits and 2-bits simplify design (e.g. alignment across cache line boundaries and memory accesses vs. 3-bits or 5-bits precision for example).
>
> 3. We had tried the scheme you mention in (3) but the results were not (as) good compared to the schemes we mention in the paper, hence we omitted this scheme from our paper.
>
> We experimented with ResNet-18 with (a) first, compressing using distillation scheme (used ResNet-34 as the teacher network) and then (b) lowered the precision to ternary mode (fine-tuning for 35 epochs with low learning rate). This experiment was done for ImageNet-1K dataset. This experiment is a variation of scheme-C in our paper where we start with full-precision networks and jointly fine-tune (use distillation scheme with warm start-up).
> Activation precision was 8-bits for this experiment. The ResNet-18 network converged to 33.13% Top-1 error rate. Comparing this with "jointly" compressing and lowering precision while training from scratch, we get 32.0% Top-1 error rate (Table-1, 4th row and 2nd column). So, our Apprentice scheme for this network and configuration is 1.13% better.
>
> Your point is well taken and we will include results where first we use knowledge distillation scheme to generate a smaller ResNet model and then lower the precision and fine-tune this small model. Currently, we have results of this scheme with ResNet-18 and few precision knobs and will collect results with this scheme for ResNet-34 and ResNet-50 for the final paper version.
> As mentioned above, the conclusions of our paper would not change and the new results will show the benefits of joint training with distillation (Apprentice scheme). Many works proposing low-precision knobs advocate for training from scratch or training with warm-startup (from weights at full-precision numerics) -- our work is in line with these observations.

---

> > ### Comment · AnonReviewer2 · 2017-12-17
> > **Upgraded review**
> >
> > Thank you for your response which has partially addressed my concerns. I think the paper quality could be improved further after the revision. So I have upgraded my review for the paper.

---

### Official Review · AnonReviewer3 · 2017-12-01
**good paper**

**Rating:** 8
**Confidence:** 4

**Review:**

Summary:
The paper presents three different methods of training a low precision student network from a teacher network using knowledge distillation.
Scheme A consists of training a high precision teacher jointly with a low precision student. Scheme B is the traditional knowledge distillation method and Scheme C uses knowledge distillation for fine-tuning a low precision student which was pretrained in high precision mode.

Review:
The paper is well written. The experiments are clear and the three different schemes provide good analytical insights.
Using scheme B  and C student model with low precision could achieve accuracy close to teacher while compressing the model.

Comments:
Tensorflow citation is missing.
Conclusion is short and a few directions for future research would have been useful.

---

> ### Author Response · Authors · 2017-12-05
> **We will do the fixes**
>
> Thank you for the reviews and comments.
>
> Missing citation: this is an oversight. We will fix this.
>
> Future directions for research:
> 1. We are currently pursuing the extension of the ideas in this paper to RNNs. Our preliminary studies on a language model for PTB dataset showed promise and based on this we are evaluating a larger data set and model like Deep Speech-2.
> 2. Some works proposing low-precision networks advocate for making the layers wider (or the model larger) to recover accuracy at low-precision. These works propose making the layers wider by 2x or 3x. While these works show the benefits of low-precision, making the model larger increases the number of raw computations. Future work could investigate low-precision and less layer widening factor (say 1.10x or 1.25x or ...). This would help inference latency while maintaining accuracy at-par with baseline full-precision.
> 3. Another interesting line of investigation for future work is looking into sparsifying networks at low-precision while maintaining baseline level accuracy and using knowledge distillation scheme during this process. As mentioned in Sec 5.5 in our paper, sparsifying a model more than a certain percentage leads to accuracy loss. Investigating hyper-sparse network models without accuracy loss using distillation based schemes is an interesting avenue of further research.

---

### Author Response · Authors · 2018-01-02
**Updated version based on reviewer feedback**

Based on the feedback we are uploading an edited version of the paper.
We added a new "Appendix section" and most of the edits are included in this section at the end of the paper.

We attempted to add more sensitivity studies on precision vs. depth and an experimental evaluation suggested by reviewer#2.
In the interest of time (and the machine resources we have), for now, these studies are done on CIFAR-10 dataset. Experimenting with CIFAR-10 allowed us to do many more ablation studies. All these studies highlight the merits of Apprentice scheme (compression and low-precision).

For the next version, we will include similar studies on ImageNet-1K dataset (we have these experiments currently running but these runs will not finish by the rebuttal deadline). We will also include a better discussion of the benefits of low-precision in terms of savings in compute and memory resources as well as the impact of these savings in resources on inference speed (reviewer#3 suggestion). We (qualitatively) discussed some of these items in section-2 but will definitely expand the discussion in the next version.

Thank you again for the feedback.

---

### Decision · Program_Chairs · 2018-01-29
**ICLR 2018 Conference Acceptance Decision**

**Decision:**

Accept (Poster)

**Comment:**

Meta score: 7

The paper combined low precision computation with different approaches to teacher-student knowledge distillation.  The experimentation is good, with good experimental analysis.  Very clearly written.  The main contribution is in the different forms of teacher-student training combined with low precision.

Pros:
 - good practical contribution
 - good experiments
 - good analysis
 - well written
Cons:
 - limited originality